# Gesture Motion Graphs for Few-Shot Speech-Driven Gesture Reenactment

Zeyu Zhao
zhaozeyu2019@ia.ac.cn
Institute of Automation, Chinese
Academy of Sciences
University of Chinese Academy of
Sciences
Beijing, China

Nan Gao
gao.nan@ia.ac.cn
Institute of Automation, Chinese
Academy of Sciences
Beijing, China

Zhi Zeng*
zhi.zeng@bupt.edu.cn
Beijing University of Posts and
Telecommunications
Beijing, China

Guixuan Zhang
guixuan.zhang@ia.ac.cn
Institute of Automation, Chinese
Academy of Sciences
Beijing, China

Jie Liu
jie.liu@ia.ac.cn
Institute of Automation, Chinese
Academy of Sciences
Beijing, China

Shuwu Zhang
shuwu.zhang@bupt.edu.cn
Beijing University of Posts and
Telecommunications
Beijing, China

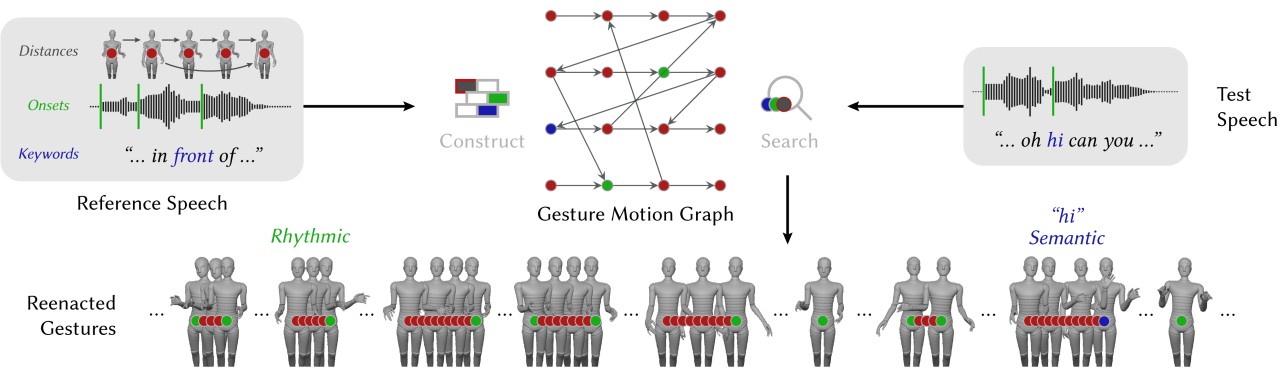

**Figure 1: Given a group of short reference speech gesture sequence, audio, and text, a gesture motion graph is constructed and ready to be searched when a group of test speech gesture audio and text is provided, for rhythmic and semantic gesture reenactment.**

## ABSTRACT

This paper presents the CASIA-GO entry to the Generation and Evaluation of Non-verbal Behaviour for Embedded Agents (GE-NEA) Challenge 2023. The system is originally designed for few-shot scenarios such as generating gestures with the style of any in-the-wild target speaker from short speech samples. Given a group of reference speech data including gesture sequences, audio, and text, it first constructs a gesture motion graph that describes the soft gesture units and interframe continuity inside the speech, which is ready to be used for new rhythmic and semantic gesture reenactment by pathfinding when test audio and text are provided. We randomly choose one clip from the training data for one test clip to simulate a few-shot scenario and provide compatible results for subjective evaluations. Despite the 0.25% average utilization of the whole training set for each clip in the test set and the 17.5% total utilization of the training set for the whole test set, the system succeeds in providing valid results and ranks in the top 1/3 in the appropriateness for agent speech evaluation.

*Corresponding author.

## CCS CONCEPTS

• **Human-centered computing** → **Human computer interaction (HCI)**; • **Computing methodologies** → Animation.

## KEYWORDS

speech-driven gesture generation, motion graph, few-shot

**ACM Reference Format:**
Zeyu Zhao, Nan Gao, Zhi Zeng, Guixuan Zhang, Jie Liu, and Shuwu Zhang. 2023. Gesture Motion Graphs for Few-Shot Speech-Driven Gesture Reenactment. In *INTERNATIONAL CONFERENCE ON MULTIMODAL INTERACTION (ICMI '23), October 9–13, 2023, Paris, France.* ACM, New York, NY, USA, 7 pages. https://doi.org/10.1145/3577190.3616118

## 1 INTRODUCTION

Generating co-speech gestures that convey rich non-verbal information remains challenging due to the indeterministic nature of the task. The one-to-many mapping between the modalities, along with other difficulties such as the lack of high-quality large-scale datasets and standardized evaluating protocols, makes it difficult to design and evaluate models for speech-driven gesture generation. In recent years, data-driven methods have attracted the interest of many researchers in the field. However, most of these methods require training on large-scale datasets. How to produce gestures in common scenarios where training data are insufficient, such as reenacting gestures with new styles naturally encoded in very few recorded gesture samples of an in-the-wild target human performer, is rarely discussed.

In this paper, we try to address this problem by designing a system that can explicitly locate key positions of rhythmic and semantic events in the sequences to form basic units of gestures and describe the continuity relationships inside. Part of that is coming from the commonly agreed observation [1, 23] that while most co-speech gestures are in synchronization with the rhythm of the voice, some gestures are more relevant to the actual meaning of the words or sentences. The other part is that it should be able to produce new gesture units that break the natural continuity relationships between units for good diversity performance. Inspired by [23], we find that motion graphs and related searching algorithms are most suitable for this task. With the gesture sequence, audio, and text of a reference speech and the audio and text of any test speech, the main idea is to construct a motion graph that describes the soft gesture units and continuity relationships inside the reference speech and search the graph for new paths of gesture frames given the test speech, as shown in Figure 1. Numerous modifications and improvements such as new pruning strategies, feature-based initialization, and fallback measures, can be made to the framework to enable compatibility with pure gesture data instead of video frames. These are proved to be the key factors for the feasibility, performance, and robustness of the system.

To gain better knowledge of how well the results produced by the system can be, we participate in this year's GENEA Challenge to evaluate our results reenacted from few-shot data and compare those with results from other systems that utilize large-scale data. To do this, we simulate a few-shot scenario by randomly choosing one clip in the provided training set as the whole reference speech for each clip in the test set, regardless of any speaker identity. For each test speech, the system only utilizes 0.25% of the whole training set on average. In such a way, the system utilizes 17.5% of the whole training set for the whole test set. Despite the low utilization of the training data, The system succeeds in producing high-quality gestures for the test set and achieves good performance in the challenge.

## 2 RELATED WORKS

Large-scale data-driven methods are becoming exceedingly popular in recent years for speech-driven data generation tasks [15], taking over rule-based methods [14] or probabilistic modeling methods [10]. Basic deep learning models show great capabilities of encoding input data and generating new gestures [3, 20]. New architectural designs that fit the specific properties of the task such as skeleton hierarchies or gesture categories are proposed to improve the performance of gesture generation [1, 13]. New generative models can also be utilized as backbones of the generation networks [19, 24].

The mixed usage of matching-based and learning-based methods can also be seen in numerous works to bypass limitations of deep learning models [4, 18]. Motion graphs are proposed to generate controllable animation from pre-recorded motion [5] and are commonly used in gesture-related tasks such as retrieval and creation [6, 16]. For speech-driven data generation, they can be utilized by defining each graph node as the feature of a sequence of gestures [22], or defining each node as a video frame [23]. Inspired by these works, we find motion graphs are suitable for our task for their inter-frame relationship description capabilities, regardless of the presence of learning-based modules. Thus, we design motion graphs for reenacting gestures from few-shot reference gesture sequences instead of large-scale data or video frames.

## 3 DATA PROCESSING

The dataset provided by the challenge organizers this year [7] is derived from the Talking With Hands data [9]. Gesture sequences, audio, text, and speaker labels of both the main agent and the interlocutor are included in the dataset, making it a dyadic dataset compared to the monadic dataset last year. As mentioned above, our system does not utilize all training data provided. Instead, we use the training set to simulate a few-shot scenario where only a small amount of data is available as reference speech. For the test set, only the audio and text data of the main agent in the test clips are utilized by the system. For each clip, only one clip in the training set is randomly chosen as the reference speech, of which only the gesture, audio, and text data of the main agent are utilized by the system. Other data including anything relevant to the interlocutor, the speaker labels, and the validation set are ignored by the system.

The data are preprocessed using the utilities provided by [2], including converting between Euler angle and exponential map rotation representation, selecting the 25 joints on upper and lower body excluding the fingers, and aligning the text to gesture frames. Since the system can work with gestures with any skeleton definition, the skeletons used inside the system are in both exponential map rotation representation and position representation. The words in the text are pre-converted to integer indices. Due to the poor quality of the hand tracking and some significant flickering on the body, we have to add 19 clips in the training set to the random selection blacklist, lock the yaw and pitch rotation of the 4 wrist-related joints, and apply the Savitzky-Golay filter with a windows length of 15 and polynomial order of 3 on the roll rotation of the 4 wrist-related joints.

# 4 METHOD

The gesture motion graph is a graph structure that can be used to represent the continuity relationships between frames in a gesture sequence regardless of the length or the skeleton definition of the sequence, as shown in Figure 2. Following [23], each node in the graph represents a frame in the gesture sequence, and each directed edge between two nodes indicates the distance between the two frames is small enough for the transition to be considered continuous. Given a reference gesture sequence and its corresponding speech audio and text, we can construct its gesture motion graph by detecting key nodes that non-uniquely split the gesture sequence into subsequences of soft gesture units and analyzing the continuity relationships between frames to find edges for unnaturally continuous frames. When we need to reenact a new test gesture sequence from its speech audio and text, we can split the test sequence into subsequences using the positions of the same kinds of key frames detected in the test speech and use a pathfinding algorithm to find the optimal paths of nodes in the graph corresponding to every test subsequence. Then a new gesture sequence that is rhythmically matched to the input speech audio and semantically relevant to the input text can be reenacted by concatenating and blending the gesture frames along the paths. Due to random operations in some fallback measures, the system may produce slightly different results at some parts for the same input.

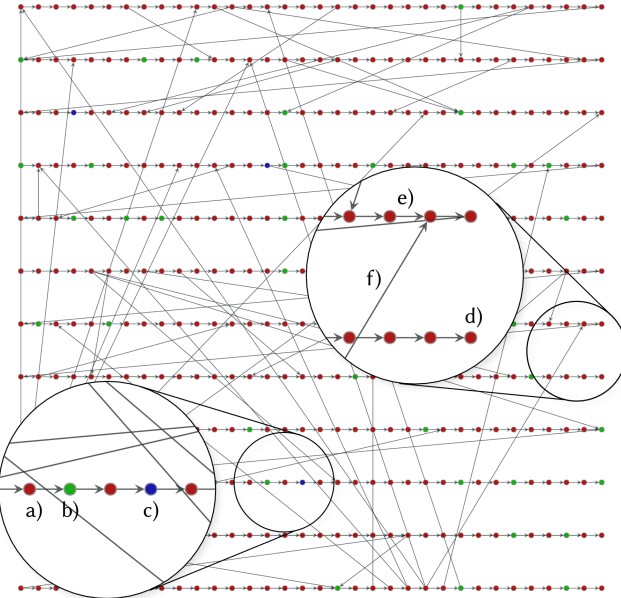

**Figure 2: A sample gesture motion graph with zoomed views of examples of a) a regular node, b) an onset node, c) a keyword node, d) a break node, e) a natural edge, and f) an unnatural edge.**

## 4.1 Graph Construction

*4.1.1 Key node detection.* After adding all frames in the gesture sequence as regular nodes into the graph, we first perform onset detection on the reference speech audio to find **onset nodes** in the gesture motion graph. The onsets are located at the backtracked peaks of the audio's spectral flux viewed as the onset strength [12], aligned to the gesture frames. Filtering on the onset strength can control the number of output onsets, which further controls the length of soft gesture units used for reenactment. Then we perform the keyword detection on the reference speech text to mark **keyword nodes** in the gesture motion graph. With the input text aligned to the frames, each word is checked to see if it belongs to a list of keywords (see [23]). If a subsequence of one or more repeating keywords is found in the text, the node corresponding to the first frame of this subsequence is then marked as a keyword node with that keyword. Also, there might be interruptions inside the speech when e.g. the speech is a composition of multiple discontinuous segments. Any frame that is not continuous with the next frame is marked as a **break node**.

*4.1.2 Continuity analysis.* We first directly add directed edges to the graph with zero weights for the frames that are naturally continuous. Then we traverse every pair of different non-continuous frames as "left" and "right" frames $\mathbf{p}_l, \mathbf{p}_r$ and calculate their distance. Here, the distance between two gesture frames, or poses, is defined to be the weighted sum of the Euclidean distance of the joint positions and the Euclidean distance of the joint velocities:

$$d_{\text{pose}}(\mathbf{p}_l, \mathbf{p}_r) = \lambda_{\text{pos}}\|\mathbf{p}_l - \mathbf{p}_r\|_2 + \lambda_{\text{vel}}\|\mathbf{v}_l - \mathbf{v}_r\|_2,$$

where the velocities $\mathbf{v}_l, \mathbf{v}_r$ can be calculated by differentiating the current and previous frames, and $\lambda_{\text{pos}}, \lambda_{\text{vel}}$ are the weights of the two terms. For every left frame, a dynamic threshold for continuity is defined to be the mean distance between the left frame and its following (up to) $l_{\text{cn}}$ frames. This threshold is used to filter out the right frames with distances that are too large to be considered continuous frames. After filtering, every remaining right frame adds a candidate directed edge to a list (not to the motion graph) with its pose distance to the left frame as the weight. However, this criterion of continuity can produce a large number of neighbored right frames for a left frame and frequently generates short loops in the graph. Thus, we perform two pruning operations to reduce the number of candidate edges. For each left frame, the first strategy is, for a continuous sequence of up to $l_{\text{pn}}$ right frames in the candidate list, we only reserve the first one and remove the others. The second strategy is, for the remaining right frames, one is removed if another edge, that starts in the $l_{\text{pn}}$-neighbor of one frame and ends in the $l_{\text{pn}}$-neighbor of the other frame, already exists in the graph. After the pruning, we add all candidate edges to the graph and move on to the next left frame.

## 4.2 Pathfinding

*4.2.1 Beam search.* The core of the path-finding algorithm is a parallelized greedy breadth-first search algorithm known as the beam search [8] for each test subsequence. Given the target path length $l_{\text{sub}}$, the termination criteria for paths, and $l_{\text{npaths}}$ initial starting nodes, the beam search algorithm outputs $l_{\text{npaths}}$ paths with top-$l_{\text{npaths}}$ minimum costs that have different lengths. These $l_{\text{npaths}}$ paths are initially one-node paths with only the given starting nodes. As shown in Figure 3, at each iteration, we initialize

an empty watch list and check if the $l_{\text{npaths}}$ paths are already terminated. All terminated paths are directly added to the watch list, and all unterminated paths are expanded by appending the children of the last node. If the last node of a path has multiple children, it should be split into multiple paths each with a child appended, which are then all added to the watch list as well. Then, we calculate the costs of all watched paths and select those with top-$l_{\text{npaths}}$ minimum costs, which are then set to be the new $l_{\text{npaths}}$ paths. Here, the cost of a path $\mathbf{P}$ is defined as the sum of the weights of the edges along the path, penalized by the difference between lengths of this path $l_{\text{path}}$ and the test subsequence $l_{\text{sub}}$:

$$c_{\text{path}}(\mathbf{P}) = \lambda_{\text{w}} \left( \sum_{i=p_1}^{p_{l_{\text{path}}-1}} w_{i,i+1} \right) + \lambda_{\text{len}} \left| 1 - \frac{l_{\text{path}}}{l_{\text{sub}}} \right|,$$

where $w_{i,j}$ is the weight of the edge $(\mathbf{p}_i, \mathbf{p}_j)$, and $\lambda_{\text{w}}, \lambda_{\text{len}}$ are the weights of the two terms. The algorithm repeats these steps and breaks when the maximum length of searching is reached or all $l_{\text{npaths}}$ paths are accepted (see appendix). Finally, the accepted path with the lowest cost is chosen for the current test subsequence.

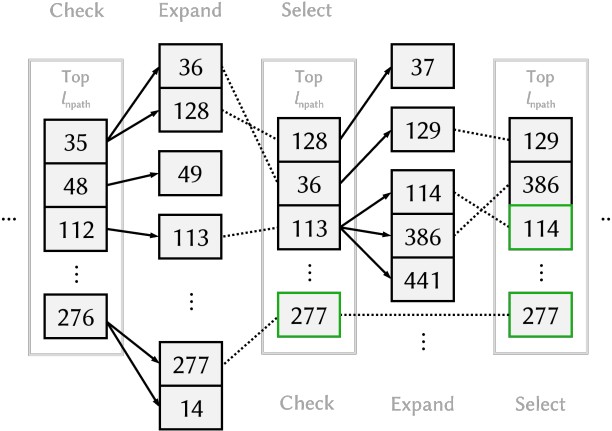

**Figure 3: An example of two iterations of the beam search process. Each iteration expands all children nodes of the last nodes of the presented paths. The expanded paths are then sorted and selected according to their costs. Terminated paths are in green.**

*4.2.2 Conditional termination.* For each test subsequence, we set the termination criteria independently based on various considerations. Normally, if the test subsequence ends at a keyword frame, the paths should terminate at any keyword node in the graph with the exact same keyword to produce semantic gestures. Otherwise, the paths should terminate at any onset or break node in the graph to produce rhythmic gestures. If no accepted path is found after the beam search is forcibly stopped, we should re-initialize the starting nodes and retry searching. Fallback measures (see appendix) can also be designed to guarantee that the beam search can stop with at least one accepted path in most cases. If no retry is needed, the beam search of the next subsequence will take the subsequent

nodes of the ending nodes as initial starting nodes, which keeps the reenacted gestures as naturally continuous as possible.

*4.2.3 Feature-based initialization.* For starting node initialization, a method based on key node features is designed for the beam search to increase the possibility of finding a path that costs less. The feature of a key node $\mathbf{f}$ is defined to be a list of lengths of the $l_{\text{feat}}$ trailing natural subsequences split by any key node, ignoring the unnatural edges:

$$\mathbf{f}_i = \{f_i - f_{i-1}, f_{i+1} - f_i, \ldots, f_{i+l_{\text{feat}}-1} - f_{i+l_{\text{feat}}-2}\},$$

where $f_j$ is the frame number of the key node with the index $1 \le j \le l_{\text{k}}$ in the ordered list of all $l_{\text{k}}$ key nodes, $f_j = 0$ when $j = 0$, and $f_j = f_{l_{\text{k}}}$ when $j > l_{\text{k}}$. For a test subsequence, we calculate the feature distance between the starting key node $k_t$ and each key node in the graph $k_m$:

$$d_{\text{feat}}(k_t, k_m) = \lambda_{\text{full}} \|\mathbf{w}_{\text{full}} \odot (\mathbf{f}_t - \mathbf{f}_m)\|_2 + \lambda_{\text{first}} \left| 1 - \frac{\mathbf{f}_{m,1}}{\mathbf{f}_{t,1}} \right| + \lambda_{\text{occ}} o_m,$$

where $\mathbf{w}_{\text{full}} \in [0, 1]^{l_{\text{feat}}}$ defines the weight for each element of the feature, $\mathbf{f}_{\cdot,1}$ represents the first element of the feature, $\odot$ is the symbol of element-wise multiplication, $o_m$ is the occurrence count of the key node $k_m$ already accepted in paths for the whole test speech, and $\lambda_{\text{full}}, \lambda_{\text{first}}, \lambda_{\text{occ}}$ are the weights of the two terms. The top-$l_{\text{npaths}}$ key nodes with minimum distances are selected to be the initial starting nodes. Fallback measures (see appendix) guarantee that there always are $l_{\text{npaths}}$ starting nodes initialized for searching after retries.

*4.2.4 Blending.* After the beam search for every test subsequence, we obtain a list of paths of pose frames in the gesture motion graph. As shown in Figure 4, we design a blending mechanism to smooth the transition between paths, as they are most likely to be discontinuous. For two paths that are needed to be concatenated, we call the last (up to) $l_{\text{blend}}$ frames of the first one left path $\mathbf{P}_{\text{l}}$ and the first (up to) $l_{\text{blend}}$ frames of the second one right path $\mathbf{P}_{\text{r}}$. We generate a path of new gestures for the concatenated left and right paths $\mathbf{P}_{\text{c}}$:

$$\begin{aligned} \mathbf{P}_{\text{c}} = (1 - \mathbf{w}_{\text{blend}}) &\odot (\mathbf{P}_{\text{l}} \oplus (\{\mathbf{P}_{\text{r},1}\} \times \min(l_{\text{r}}, l_{\text{blend}}))) \\ + \mathbf{w}_{\text{blend}} &\odot ((\{\mathbf{P}_{\text{l},l_{\text{l}}}\} \times \min(l_{\text{l}}, l_{\text{blend}})) \oplus \mathbf{P}_{\text{r}}), \end{aligned}$$

where $\mathbf{w}_{\text{blend}}$ is the weight vector, $\oplus$ is the symbol of concatenation, $\times$ is the symbol of repeating all elements in a vector, $\mathbf{P}_{\cdot,i}$ is the $i$-th node in a path, and $l_{\text{l}}, l_{\text{r}}$ are the lengths of left and right paths. The weight vector can be generated by linear, sigmoid, or other functions that map evenly-placed values to the range of $(0, 1)$. For skeletons defined as exponential map rotations of the joints, we can also convert those to quaternions and use spherical linear interpolation (SLERP) to blend the rotations, instead of using direct weighted sum.

## 5 EVALUATIONS

To evaluate the effectiveness of the system, we generate results using the mentioned data and method with the following configuration: $\lambda_{\text{pos}} = \lambda_{\text{vel}} = 1$, $l_{\text{cn}} = 5$, $l_{\text{pn}} = 10$, $l_{\text{npaths}} = 20$, $\lambda_{\text{w}} = \lambda_{\text{len}} = 1$, $l_{\text{feat}} = 10$, $\lambda_{\text{full}} = \lambda_{\text{first}} = 1$, $\lambda_{\text{occ}} = 0.5$, $\mathbf{w}_{\text{full}} = \{1, 0.5, 0.5, 0.2, 0.2, 0.2, 0.1, 0.1, 0.1, 0.1\}$, and minimum onset strength threshold 5.

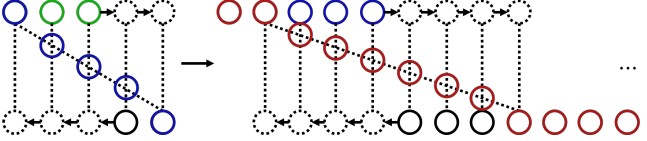

**Figure 4: An example of the blending process. The green and black paths are blended to form a blue path (left), which is then blended with another black path to form a red path (right).**

## 5.1 Subjective Evaluation

The generated results in Euler angle rotation representation (converted from exponential map) are submitted to the challenge organizers and evaluated by the human evaluators recruited from six English-speaking countries [7]. Three aspects of the generated results are evaluated and released to the participants, including the human-likeness, the appropriateness for agent speech, and the appropriateness for the interlocutor. We do not discuss the last one since it assumes that the systems are interlocutor aware, which is not the case for our system. No objective evaluation result is available to the participants. Videos used in this evaluation are available at https://zenodo.org/record/8211449.

*5.1.1 Appropriateness for agent speech evaluation.* As mentioned, to simulate a few-shot scenario, for each test clip (minimum 60 seconds, maximum 77 seconds, 62.4 seconds on average), only one training clip is randomly chosen as the reference speech. For the 70 given test clips, 70 different training clips are finally chosen. Each chosen training clip (minimum 60 seconds, maximum 427 seconds, 170.2 seconds on average) only constitutes a tiny portion (minimum 0.088%, maximum 0.627%, 0.25% on average) of the whole training set (68069.9 seconds). For the whole test set, only 17.5% of the training data are utilized to produce the results. Despite the low utilization of the training set, the results generated by our system (labeled **SK**) got a good mean appropriateness score (MAS) of $0.18 \pm 0.06$, ranking fourth among the 12 participants (top 1/3). The full results can be found in Table 1 and Figure 5. This shows that the system is able to produce high-quality results that are comparable with systems utilizing large-scale datasets.

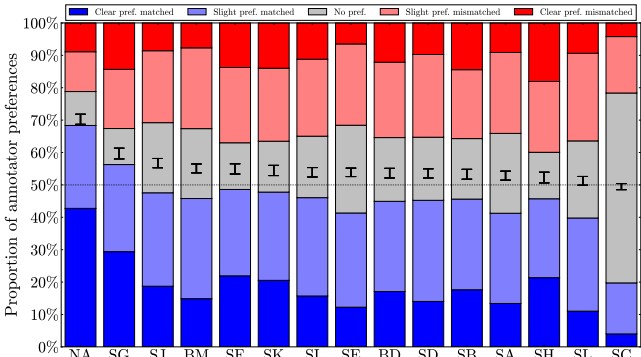

**Figure 5: Bar plots visualising the response distribution in the appropriateness for agent speech study [7].**

**Table 1: Appropriateness for agent speech [7]**

| Condi-tion | MAS | Pref. matched | Raw response count | | | | | |
|---|---|---|---|---|---|---|---|---|
| | | | 2 | 1 | 0 | −1 | −2 | Sum |
| NA | 0.81±0.06 | 73.6% | 755 | 452 | 185 | 217 | 157 | 1766 |
| SG | 0.39±0.07 | 61.8% | 531 | 486 | 201 | 330 | 259 | 1807 |
| SJ | 0.27±0.06 | 58.4% | 338 | 521 | 391 | 401 | 155 | 1806 |
| BM | 0.20±0.05 | 56.6% | 269 | 559 | 390 | 451 | 139 | 1808 |
| SF | 0.20±0.06 | 55.8% | 397 | 483 | 261 | 421 | 249 | 1811 |
| SK | 0.18±0.06 | 55.6% | 370 | 491 | 283 | 406 | 252 | 1802 |
| SI | 0.16±0.06 | 55.5% | 283 | 547 | 342 | 428 | 202 | 1802 |
| SE | 0.16±0.05 | 54.9% | 221 | 525 | 489 | 453 | 117 | 1805 |
| BD | 0.14±0.06 | 54.8% | 310 | 505 | 357 | 422 | 220 | 1814 |
| SD | 0.14±0.06 | 55.0% | 252 | 561 | 350 | 459 | 175 | 1797 |
| SB | 0.13±0.06 | 55.0% | 320 | 508 | 339 | 386 | 262 | 1815 |
| SA | 0.11±0.06 | 53.6% | 238 | 495 | 438 | 444 | 162 | 1777 |
| SH | 0.09±0.07 | 52.9% | 384 | 438 | 258 | 393 | 325 | 1798 |
| SL | 0.05±0.05 | 51.7% | 200 | 522 | 432 | 491 | 170 | 1815 |
| SC | −0.02±0.04 | 49.1% | 72 | 284 | 1057 | 314 | 76 | 1803 |

*5.1.2 Human-likeness.* However, our system did not get a satisfying median score (37 ∈ [35, 40]) in the human-likeness evaluation, ranking ninth among the 12 participants. Since our system reenacts new gestures from the raw gesture frames of the reference gesture sequence, the quality of the results is heavily affected by the quality and the length of the reference data. Flickering or other defects existing in the naturally continuous frames and the lower-than-needed training data utilization can be possible causes of the low ratings given by the evaluators. Also, the blending process can only guarantee smooth transitions between paths. If too many transitions occur in a very short time span, it may give the evaluators some non-humanlike impression. In a word, increasing the quality of the reference speech data and using more training data as reference speeches may give a better score in this evaluation.

## 5.2 Ablation Study

Pruning strategies, feature-based initialization, fallback measures, and other new designs for the gesture motion graph are key factors for the feasibility, performance, and robustness of the system. To justify this, we also conduct ablation studies using the results in joint position representation. We evaluate our system in three setups on three objective metrics. The **weak detection** setup removes proper filtering measures in onset detection (with minimum onset strength threshold 0). The **weak pruning** setup degrades pruning operations in continuity analysis ($l_{pn} = 1$). The **weak initialization** setup initializes random starting nodes in the beam search algorithm. The first metric is for **motion synchronization (Syn)** [17], which calculates the differences between velocity magnitudes of the generated and ground truth gestures at each frame. Note that the results of such distance comparisons cannot accurately measure the quality of the generated gestures. The second metric is a score for **beat consistency (BC)** [11] that measures the beat correlation between gestures and speech audio by calculating the mean distance between the audio onsets and the nearest

**Table 2: Ablation study results**

| Setup | Syn↓ | BC↑ | Div↑ | #Failure |
|---|---|---|---|---|
| Weak Detection | 0.61393 | 0.021577 | 0.06101 | 0 |
| Weak Pruning | 0.57947 | **0.022278** | 0.05795 | 0 |
| Weak Initialization | 0.58290 | 0.021982 | 0.06639 | 0 |
| No Term. Fallback | - | - | - | 51 |
| Full | **0.57866** | 0.022087 | **0.07461** | 0 |

peaks of angle change rate. The third metric is for **gesture diversity (Div)** [21]. It calculates the ratio of large angular changes of velocities between frames and uses that to indicate the frequency of motion changes. Finally, another **no termination fallback** setup that disables all termination fallback measures is added and the **number of failures** (stuck in infinite loops) during pathfinding is counted to demonstrate the necessity of these measures. We see in Table 2 that although weak setups sometimes produce gestures with a better rhythmic score, they perform much worse in velocity similarity to ground truth or gesture diversity. Moreover, the system fails 51 times out of 70 (73%) without the fallback measures, showing that these designs are necessary for the graph to work with few-shot gesture data.

## 6 CONCLUSION

In this work, we propose a system for reenacting gestures in few-shot scenarios where very few reference samples are available based on gesture motion graphs. The input reference gesture and speech data are analyzed and a gesture motion graph with descriptions of the interframe continuity and key rhythmic and semantic events is constructed. Given the test speech, a path of blended pose frames can be searched from the gesture motion graph to form a new sequence of reenacted gestures. The evaluations show that the system can generate high-quality results comparable with methods designed for large-scale data, and the new designs succeed in providing robust performance for the system.

Nevertheless, this system has its limitations in multiple aspects. For example, although the requirement for data size is reduced, the reference data still need to be high quality for reenactment. Also, the construction and search processes are manually designed based on human prior knowledge with some of the thresholds that need to be tuned manually. We can explore learning-based methods that can enhance the mechanisms of key node detection, path cost, etc.

## ACKNOWLEDGMENTS

This work was supported by the National Key R&D Program of China (2022YFF0901902).

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

# A  METHOD DETAILS

## A.1  Pathfinding

*A.1.1  Termination Fallback Measures.* If no matching keyword node is found after multiple retries, the stopping nodes should fall back on onset or break nodes. Also, if the shortest subsequence in the graph is still much longer than the target length, it is difficult for any output path to be considered accepted, in which case the number of retries keeps increasing endlessly. This can be solved by randomly discarding some stopping nodes gradually to destructively lengthen the subsequences. If this operation does not stop the number of retries from endlessly increasing, that means the longest subsequence is still much shorter than the target length. In this case,

we can return a path with the target length by repeating the last node of the previous search result and terminating the search for the current test subsequence.

*A.1.2  Path Acceptance.* A path is considered accepted when it is terminated and 0.9 to 1.1 times the length of the test subsequence. The accepted path with the minimum cost is selected to be the search result if any exists, which is then resampled evenly if the length of this path $l_{\text{path}}$ is not equal to the target length $l_{\text{sub}}$.

*A.1.3  Initialization Fallback Measures.* On each retry, the last top key nodes are discarded and the next top-$l_{\text{npaths}}$ key nodes are selected. If no sufficient key node is available, we can randomly select $l_{\text{npaths}}$ arbitrary nodes as a fallback measure.