# OpenReview forum: "Gesture Motion Graphs for Few-Shot Speech-Driven Gesture Reenactment"
_ACM.org/ICMI/2023/Workshop/GENEA_Challenge — GENEA Challenge 2023 Mainproceeding_

### Official Review · Reviewer_spEZ · 2023-07-29
**The paper proposed a few-shot method based on motion graph. The idea is well motivated with the clear exposition that allows replication. Overall a solid paper that will be of interest for the workshop and challenge attendees**

**Rating:** 8
**Confidence:** 5

**Review:**

## Paper Summary

The paper proposed a co-speech gesture synthesis method based on motion matching. Specifically, it builds a gesture motion graph by detecting key nodes that split the motion sequences into multiple sub-segments. A transition edge going from one node to another is also identified based on similarity distance between frames. To avoid excessive nodes and edges, two pruning strategies are utilized to reserve only an important subset of edges. The motion synthesis is done by doing a beam search within the built motion graph to find best candidate path that matches the input speech.

## Strength

The proposed method is well-motivated and the method design are discussed in details for reproducibility. While motion graph has been applied in various applications before, it is interesting to see the method based on motion matching being utilized in gesture synthesis. The method allows using only a subset of training motions to achieve competent results. By using explicit motion matching for synthesis, the method is also able to offer more clarity from the evaluation results about what works or not (i.e. higher appropriateness with keyword nodes in the graph).

## Weakness

Some paragraphs may need more details or clarifications. For example, in Line 271, the definition for discontinuous frames is not well discussed. Similarly, in Section 4.2.3, it may be helpful to have a formal definition for the feature $f$ for a key node.

Would be helpful to discuss further about the few-shot property of the proposed method. Specifically, the reviewer is interested to learn about the trade-off between the data utilization and  the quality of gesture synthesis. For example, how feasible is the method to utilize only X minutes of gesture motions and still produce reasonable results.

## Rating Justification

Overall the paper proposed a new method based on motion graph. The main advantage is the few-shot learning, which only requires a smaller subset of training data to achieve competent results. The exposition is clear and includes interesting ideas for replications and future improvements. I believe the paper will be helpful for the workshop and challenge attendees.

---

### Official Review · Reviewer_qWj2 · 2023-07-30
**A motion graph-based gesture generation system, which could replicate the unique gestural style of a speaker in the wild based on a few speech samples. I rate this paper as Borderline Accept.**

**Rating:** 6
**Confidence:** 5

**Review:**

This work introduces a system that produces relevant gestures based on provided text and speech. It is particularly effective in scenarios where data is limited, such as replicating the unique gestural style of a speaker in the wild based on a few speech samples. Given a reference speech clip that includes gestures, audio, and text, the system first creates a gesture motion graph. It then applies a pathfinding algorithm to reenact rhythmically and semantically appropriate gestures for the test data, which comprises both audio and text.

This study presents a fresh approach to tackle the data scarcity issue prevalent in the field of co-speech gesture generation. It showcases that traditional motion graphs can still deliver satisfactory results in gesture generation tasks. However, this article is somewhat condensed and the lack of extensive details has left me with many unanswered questions:

1. Section 4.1.1 references the use of keyword detection to establish keyword nodes for the reference speech text. I want to know how the keyword collection is formulated, and how do we ensure that this collection is scientifically valid at the semantic level?

2. In Section 4.2.2, it is detailed that for a test subsequence, if an onset or a semantic keyword is present in its end frame, then the system will take explicit control to determine whether the node corresponding to the generated gestures should be an onset node or a keyword node. However, the article fails to mention what happens with the onset and keyword at the subsequence's other frames (beginning or middle). This oversight seems unreasonable.

3. In Section 5.1.1, it's noted that for the provided 70 test samples, the authors selected 70 samples from the training data as the reference speech for action graph generation. What evaluation metrics guided this selection process? For test data beyond the provided samples, could an automated selection algorithm be developed to choose the most appropriate sample from the given data repository to construct the motion graph?

In addition, there are some minor issues as below:

- In line 394 of the paper, the first word in the phrase "or the reference sequence" should be of.

**Summary:**

- **Main Pros:**
  - This approach offers a solution for generating lively gestures when there the data is insufficient.

  - The concept of consciously controlling the rhythmic or semantic aspects of the generated gestures, by categorizing the nodes in the motion graph, is insightful.

- **Main Cons:**
  - Gesture generation systems based on motion graphs tend to be less adaptable, demand high-quality and diverse reference speech data, and struggle with generating long sequences of gestures.

  - Some key parts of the article are vaguely explained.

- **Rating: 6(Marginally above acceptance threshold)**
  - I assigned this score because, as noted earlier, this work achieves good results with small datasets, a success that's encouraging to witness.

---

### Decision · Program_Chairs · 2023-08-04

**Decision:**

Accept (Main proceeding)

**Comment:**

This work is an important submission to the GENEA challenge, as it utilises motion graphs instead of deep learning. The core idea is to construct motion graphs from gesture sequences, where nodes are “interesting” poses (with quick movement or an associated audio onset/keyword) and weighted edges denote the distances between poses.

As noted by reviewer qWj2, the paper lacks information about how the nodes are selected to match the speech. My understanding, mostly based on [23], is the following:
- The input audio sequence is split into subsequences separated by audio onsets and keywords.
- The first subsequence S starts from a random node, then 1) if S ends on a keyword, the algorithm looks for a path that ends on a node with that keyword; 2) if S ends on an audio onset, then the algorithm looks for a path that ends on an audio onset or a “break” node (corresponding to quick movement).
- The remaining subsequences are constructed the same way, starting from the selected ending node of the previous subsequence.
- (The subsequences are then blended together following Section 4.2.4.)

Overall, all reviewers agree that this is a mostly well-written paper with a considerable level of original contribution. I recommend acceptance to the ICMI main proceedings, with the following requests for the camera-ready version:
- Please revise the text so that the method can be understood without referring to [23].
   * To this end, I strongly suggest adding pseudocode to the paper that describes the node selection process (and other parts of the method, e.g., initialisation).
- A formal description of the node features f, as the current text is somewhat unclear (suggested by reviewer spEZ)
- Provide qualitative results, for example, by linking to a video.

Additionally, I gently suggest:
- Explicitly stating whether the definition of discontinuity on line 272 is the same as in the following section (suggested by reviewer qWj2)
- Explicitly stating that the list of keywords are the same as in the appendix of [23]
- Briefly comparing the severity of artefacts introduced by discontinuous frames within a single path and blending separate subsequences (in other words, which of the two have more effect on the?)
- Commenting on computational requirements of the method - in particular, would it be feasible to construct a single graph instead of 70 different ones?